# Differential Antinociceptive Efficacy of Peel Extracts and Lyophilized Juices of Three Varieties of Mexican Pomegranate (*Punica granatum* L.) in the Formalin Test

**DOI:** 10.3390/plants12010131

**Published:** 2022-12-27

**Authors:** José Antonio Guerrero-Solano, Mirandeli Bautista, Josué Vidal Espinosa-Juárez, Luis Alfonso Moreno-Rocha, Gabriel Betanzos-Cabrera, Liana Claudia Salanță, Minarda De la O Arciniega, Elena G. Olvera-Hernández, Osmar Antonio Jaramillo-Morales

**Affiliations:** 1Institute of Health Sciences, Academic Area of Nursing, Autonomous University of the State of Hidalgo, Circuito Ex Hacienda La Concepción S/N Carretera Pachuca Actopan, San Agustín Tlaxiaca, Hidalgo 42160, Mexico; 2Institute of Health Sciences, Academic Area of Pharmacy, Autonomous University of the State of Hidalgo, Circuito Ex Hacienda La Concepción S/N Carretera Pachuca Actopan, San Agustín Tlaxiaca, Hidalgo 42160, Mexico; 3School of Chemical Sciences, Autonomous University of Chiapas, Carretera Panamericana Km. 2.5 S/N, Ocozocoautla de Espinosa, Chiapas 29120, Mexico; 4Biological Systems Department, Autonomous Metropolitan University, Xochimilco Unit, Calzada del Hueso 1110, Villa Quietud, Coyoacán, Mexico City CDMX 04960, Mexico; 5Institute of Health Sciences, Academic Area of Nutrition, Autonomous University of the State of Hidalgo, Circuito Ex Hacienda La Concepción S/N Carretera Pachuca Actopan, San Agustín Tlaxiaca, Hidalgo 42160, Mexico; 6Faculty of Food Science, University of Agricultural Sciences and Veterinary Medicine Cluj-Napoca, Mănăştur 3-5, 400372 Cluj-Napoca, Romania; 7Life Sciences Division, Nursing and Obstetrics Department, Campus Irapuato-Salamanca, University of Guanajuato, Ex Hacienda el Copal, Km. 9 Carretera Irapuato-Silao, A.P. 311, Irapuato 36500, Mexico

**Keywords:** antinociceptive, pomegranate, *Punica granatum* L., peel, juice, formalin test, nociceptive pain, inflammatory pain

## Abstract

Pharmacological treatment of pain often causes undesirable effects, so it is necessary to look for natural, safe, and effective alternatives to alleviate painful behavior. In this context, it is known that different parts of pomegranate have been widely consumed and used as preventive and therapeutic agents since ancient times. For example, it has been shown to have an antinociceptive effect, however, there are many varieties. Each part has been found to display unique and attractive pharmacological activities. The content of the active phytochemicals in pomegranate depends on the cultivar, geographical region, the maturity, and the processing method. In this context, the effects of various pomegranate varieties and other parts of the pomegranate (e.g., peel and juice) on pain behavior have not been examined. The aim was to evaluate and compare the antinociceptive effect of ethanolic extracts (PEx) and lyophilized juices (Lj) of three varieties of pomegranate in the formalin test. In addition, computer-aided analysis was performed for determining biological effects and toxicity. Peels were extracted with ethanol and evaporated by rotary evaporation, and juices were filtered and lyophilized. Wistar rats (*N* = 48) were randomly distributed into 8 groups (*n* = 6) (Vehicle, Acetylsalicylic Acid, PEx1, PEx2, PEx3, Lj1, Lj2, and Lj3). The formalin test (2%) was carried out, which consists of administering formalin in paw and counting the paw flinches for 1 h, with prior administration of treatments. All samples have an antinociceptive effect (phase 1: 2.8–10%; phase 2: 23.2–45.2%). PEx2 and Lj2 had the greatest antinociceptive effect (57.8–58.9%), and bioactive compounds such as tannins and flavonoids showed promising pharmacodynamic properties that may be involved in the antinociceptive effect, and can be considered as a natural alternative for the treatment of nociceptive and inflammatory pain.

## 1. Introduction

Pain is defined as “an unpleasant sensory and emotional experience associated with, or resembling that associated with, actual or potential tissue damage” [1]. According to the most widely accepted classification, there are four types of pain: nociceptive, inflammatory, neuropathic, and functional [2]. Pain normally occurs in response to the activation of nociceptors (nociceptive pain), alerting to the presence of potentially noxious stimuli, which assume a protective role. However, clinical pain can arise from chronic inflammatory states (inflammatory pain), such as secondary to nervous system damage (neuro-pathic pain), even when there are no signs of damage that may cause it (dysfunctional pain) [2,3]. There are several preclinical models for the study of pain [4]. For example, for peripheral pain, the most relevant models consist of generating a mechanical, thermal, or chemical injury (by injecting chemical agents subcutaneously, muscle, joint, or visceral), among other manipulations that replicate pain situations in specific conditions [5]. One of the tests based on chemical injury is the formalin test. In this test, 30 min after the administration of the experimental drug, the number of paw flinches or paw licks is measured after a subcutaneous injection of formalin in the dorsal surface of the right hind paw, usually for 60 min. This test has a biphasic response and is useful for assessing nociceptive (first 10 min of the test) and inflammatory (15 to 60 min) pain, where acute or short-term inflammation is generated [6]. In general, the formalin test is sensitive to nonsteroidal anti-inflammatory drugs (NSAIDs), and mild analgesics at doses relevant to acute pain in general [5].

Despite being considered beneficial and vital for survival, pain, whether acute or chronic, generates aversion and, in many cases, suffering in those who must endure it [7]. This motivates further efforts to develop drugs to modulate neural activity and reduce or eradicate unpleasant pain. In this sense, there are many types of drugs and adjuvants for the treatment of pain [8], however, many of these drugs have undesirable effects, as in the case of non-steroidal anti-inflammatory drugs (NSAIDs), such as acetylsalicylic acid (ASA), which frequently cause gastric ulceration, gastritis, and to a lesser extent renal, hepatic, and cardiac disease [9,10].

Therefore, in recent years there has been a growing interest in replacing or reducing the undesirable effects of drugs with natural compounds, such as herbal extracts, dietary supplements, or isolated phytochemicals. These are considered safe, effective, and with or without minor side effects [11]. Many of these natural sources have been used since ancient times in traditional medicine to reduce pain and inflammation [12].

Pomegranate (*Punica granatum* L.) is one such natural source used since ancient times as a food, juice, and as a functional food for medicinal purposes. It is a tree whose origin is located in the Middle East (in the region where Iran, Afghanistan, and North India are currently located) [13], although it is currently cultivated in all regions of the world where climatic conditions and salinity allow it [14]. Over the years, the flowers, leaves, roots, bark, and fruit of pomegranate have been used in many regions of the planet; in traditional medicine it has been used as a taenicide and antiparasitic, antimicrobial, antidiarrheal, astringent, antihemorrhagic, as a healing agent, for burns, as a powerful anti-inflammatory, and for pain [15]. It is this last therapeutic effect that interests us, since in traditional Mexican medicine it is widely used for this purpose [16]. Current science has tried to verify the mechanisms of action of these effects, finding sufficient evidence that different parts of the pomegranate have an antinociceptive (analgesic) effect in preclinical and clinical models [17]. Pomegranate peel and juice have been shown to have a good antinociceptive effect, which is attributed to the presence of polyphenols such as ellagitannins (and their metabolites), anthocyanins, free organic acids, terpenes, and alkaloids [18]. However, it is estimated that there are approximately 500 varieties of *Punica granatum* L., and the composition of phytochemical substances in plants is very varied, both from a qualitative and quantitative point of view. This variability may even exist between different varieties of the same product, such as the pomegranate, which could, in this way, have different beneficial properties for health [19]. Therefore, this study aims to evaluate and compare the antinociceptive effect of peel extracts and lyophilized juice of three Mexican pomegranate varieties in the formalin test and to determine the biological effects and toxicity prediction of its compounds.

## 2. Results

Formalin (2%) was administered subcutaneously in the right hind paw of rats to elicit nociceptive behavior. Phase one was assessed from minute zero to minute ten, and phase two, inflammatory response, was assessed from minute 15 to minute 60.

The time course showed a typical biphasic response observable in the Vehicle group. Figure 1a,b show the time course of the flinches per minute of the different groups studied (PEx1, PEx2, and PEx3, as well as Lj1, Lj2, and Lj3), compared to the groups receiving vehicle and ASA. In Figure 1a,b it can be seen that the Vehicle group shows a typical behavior of the test, and for both ASA and treatments, the number of flinches was reduced compared to Vehicle, finding a greater reduction in nociceptive behavior in PEx2 and Lj2 (Figure 1).

Afterwards, the area under the curve (AUC) was analyzed by phases and overall, for all groups by the trapezoid method and from it, the percentage of antinociception was calculated. Figure 2 shows the comparison by groups of the percentage of antinociception, by phases (phase 1 and phase 2). It was found that all peel extracts and lyophilized juices, as well as ASA, have an antinociceptive effect in both phases of the formalin test, having a greater effect in phase 2 (25 to 50% antinociception). All peel extracts and lyophilized juices had a statistically significant difference (*p* < 0.05) versus Vehicle group, and in phase 1, the PEx2 group had a positive statistically significant difference versus ASA.

Figure 3 shows the group comparison of overall antinociception (both phases). All peel extracts and lyophilized juices, as well as ASA, were found to have an overall antinociceptive effect, with a greater effect on the green variety (both PEx2 and Lj2). All the peel extracts and lyophilized juices had a statistically significant difference (*p* < 0.05) versus Vehicle group. It should be noted that Lj2 had a similar effect to PEx2, but both have no statistically significant difference versus ASA.

The results of the in silico analysis involving the biological activities of the components are reported in *Punica granatum* L. The results are displayed as Pa, the probability that the molecule has that pharmacological activity, or pi, the probability that the molecule does not have the activity, where 1 represents 100% probability that the event will take place. It is clear that compounds called tannins, such as punicalagin, have a high probability of generating anti-inflammatory and antioxidant effects, while flavonoids also tend to show antinociceptive activity (Table 1). To make the analysis more complete, it was decided to identify the mechanisms associated with anti-inflammatory, antinociceptive and antioxidant effects (Table 2). Here we can observe that the effect of tannins could be generated by the inhibition of kinases such as PKA or PKC, while for flavonoids their effect could be associated with the inhibition of inducible nitric oxide synthase (iNOS), inhibition of the release of histamine, or to the decrease in the expression of some cytokines such as tumor necrosis factor.

The therapeutic potential of compounds derived from plants can be compromised by deficiencies in absorption and distribution, and these in turn depend on the physicochemical characteristics of each molecule, which is why we analyze these parameters in silico (Table 3). LogP is a parameter that defines the partition coefficient and allows for determining the behavior of a molecule in the environment of biological fluids. It is a parameter that measures the hydrophobicity of the molecule, and the reference parameter is ≤5; TPSA refers to a molecular descriptor that takes into account the molecular surface area of polar atoms such as oxygen, nitrogen, and their hydrogens attached to these atoms. Most of the molecules that present good oral absorption are between 100–150 Å, while to cross the blood–brain barrier, ≤90 Å is required. The molecular weight (MW) is related to the size of the molecule; the larger the molecule, the larger the water cavity must be that is generated to solubilize the compound, and passive diffusion is also complicated. An adequate molecular weight would be ≤500 g/mole. Another crucial process is hydrogen bonds, which increase aqueous solubility, and must be broken for the molecule to cross the plasma membrane. This characteristic is determined by the number of hydrogen donors (HBD), that is, the sum of hydrogen hydroxyl groups (OH) and amino aminos (NH) in the molecule and by the number of hydrogen acceptors (HBA) that are obtained from the sum of oxygen and nitrogen in the molecule. The number of rotatable bonds (RB) refers to a topological parameter that identifies the flexibility of the molecule, and it has been described that the appropriate number must be less than 12. Together, these parameters allow us to predict the oral bioavailability of a substance, and from these descriptors the Lipinski rule or rule of 5 arose, which mentions that if a molecule does not meet two or more of these characteristics, it will have permeability problems in biological membranes. In this sense, we can note that most of the molecules are within the range and do not present violations, since most of these descriptors are within the standard ranges.

Additionally, the table shows undetected pan-assay interference components (PAINS), since it is very common for some molecules to present positive effects in silico assays. However, when they are translated to in vivo effects, these do not occur. This is due to small reactive or otherwise susceptible molecules that are contained as substructures in larger compounds. In that sense, it can be noted that most molecules do not present these alerts, which is very positive.

The data in Table 4 complement the physicochemical parameters and show that a large part of the molecules present good bioavailability by the oral route, however, only some of them would reach the central nervous system because they easily cross the blood–brain barrier.

Finally, the data from the in silico analysis of the toxicity predictions of *Punica granatum* L. components are shown in Table 5. Two platforms were used for this analysis, and it can be seen that most of the molecules do not present a considerable risk in parameters such as mutagenicity (AMES toxicity), or blockage of the human ether-related gene (hERG) which is known to cause cardiacal alterations, including the mean lethal doses (LD_50_) for most compounds found in categories II (values greater than 50 mg/kg but less than 500 mg/kg) or III (values greater than 500 mg/kg but less than 5000 mg/kg).

## 3. Discussion

The aim of this study was to evaluate and compare the antinociceptive effect of three Mexican varieties of pomegranate (peel extract and lyophilized juice of bittersweet, green and red pomegranate) in the formalin test, a model of acute persistent pain, and to determine the biological effects and toxicity prediction of its compounds. For this purpose, two phases are involved (phase 1, 0 to 10 min and phase 2, from 15 to 60 min), with an interface between them (10–15 min). In the first phase, following a nociceptive stimulus, activation of neurons in the dorsal horn of the spinal cord was involved, with an increase in the activity of C and Aδ fibers, as well as the release of substance P. In the second phase, there is a release of inflammatory mediators such as prostaglandins, nitric oxide, ATP, histamine, and bradykinin, among other molecules [20].

At a dose of 316 mg/kg, it was found that both peels and juices decreased the nociceptive behavior stimulated by formalin administration. In this regard, a weak antinociceptive effect of peel extracts and lyophilized juices was observed in the first phase of the test, however PEx2 obtained the maximum percentage found for this phase (10.6%) with statistically significant difference versus the ASA and Vehicle group.

Some alkaloids are known to have an effect in both phases of the formalin test [21]. In this context, we think that the short effect in phase 1 can be explained because pomegranate peel and pomegranate juice have some alkaloids like caffeine, peelletierine, and punigranate (a pyrrolidine type alkaloid), although they are not considered rich in these compounds [22,23,24]. Previous reports indicate that pomegranate peel of the Sefri variety has pain inhibition percentages (IPI) like ours (4 to 8.9%), but not the peel of the Amrouz variety, which achieved inhibition percentages of 37 to 75% (at doses of 25, 50, and 100 mg/kg intraperitoneal weight before injection of 20 µL of 2% formalin) [25].

Centrally acting drugs (such as opioids) inhibit both phases of the formalin test, as their mechanism of action involves effects produced by endogenous prostaglandins and opioids [26]. However, this is not the case with peripherally acting drugs, such as ASA, since these reduce nociception mainly in the second phase by inhibiting inflammatory mechanisms [27,28]. This explains the brief effect of ASA in the first phase of the test in this study. The data suggest that the peel and juice extracts exert their analgesic action slightly at the central level, and largely at the peripheral level.

Likewise, in phase 2 of the test, it was found that the juice of the same variety, Lj2, obtained the highest percentage of antinociception, with 48.6%, followed by PEx2 with 44.78% and PEx3 with 44.55%. It is likely that the green peel variety has a higher amount of phytochemicals than the other two, both in the peel and in the juice; it has been proven by efficient HPLC methods that certain varieties have higher concentrations of gallic acid, punicalagins, ellagic acid, among others, in pericarp, juice, seed, and pomegranate leaves, or it may be a consequence of the content and composition of effective components, i.e., the active phytochemicals in medicinal plants vary with changes in the growth seasons, growth years environment, cultivar, geographical region, the maturity, and the processing method [29]. There are antecedents in which the antinociceptive effect of methanolic and hydroalcoholic extracts was evaluated in the formalin test in rats and mice. Unlike this study, in these, the route of administration was intraperitoneal, intracerebrovascular, and topical, at doses ranging from 10, 25, 50, 150, and up to 400 mg/kg weight. The greatest effect was found to occur in phase 2 of the formalin test [25,30,31], which supports the present result and compels consideration of further evaluation of the analgesic capacity of the extract with the greatest effect.

Another interesting result is that there is no dose-dependent effect on Lj. It is important to note that since there is a wide variety of components, in some cases, positive interactions can occur, generating potentiation. However, negative interactions can also manifest, such as antagonism; one possibility with Lj2 is that metabolites are present in the correct quantity that generate summation effects with respect to Lj1, however, by continuing to increase the doses as in Lj3. It is possible that the necessary amount of a metabolite that blocks or counteracts the antinociceptive effect is reached, and it would be important to validate this result in the future. Regarding the mechanisms of action of the effect of pomegranate peel and juice, it has been documented that some of its bioactive compounds act as follows: (a) Antioxidants, in general, stabilize nitric oxide (NO) and prolong its cellular concentration, protecting it from free radicals. These data are in agreement with the in silico analysis performed during this research. (b) Some *Punica granatum* L. molecules act as inhibitors of PKA or PKC, which have been shown to contribute to analgesic activity, since these enzymes are involved in the induction of the expression of inflammatory mediators such as prostaglandins and histamine [32]. (c) Gallic acid acts as an antagonist of transient receptor potential ankyrin 1 (TRPA1) channels. This channel plays an integral role in pain and neurogenic inflammation through activation of sensory nerves both centrally and peripherally. (d) The tannins punicalagin, punicalin, strictin A, and granatin B reduce IL-6 expression and may be responsible for the modulation of the NF-κB pathway. They inhibit NO production, prostaglandin E2 (PGE2), and COX-2 expression. (e) Some metabolites of tannins (urolithin A, glucuronide and their aglycones) decrease IL-8 expression. (f) Corilagin significantly reduces capsaicin-induced nociception, suggesting that this tannin may be involved in TRPV1 channel antagonism. (g) Some flavonoids present in pomegranate inhibit COX-2 activity and expression and PGE2 production. (h) Ellagic acid increases nitric oxide synthase (NOS) levels, increases local NO generation, inhibits COX-1 and COX-2, and activates the opioid system [33,34,35,36,37,38]; it is interesting that these approaches to corroborate the results are carried out in in vitro or in vivo studies in the future.

This paper focuses on the analgesic study of products derived from plants that are administered orally, so a good bioavailability of the metabolites by this route of administration is necessary. In this sense, we carry out in silico studies to verify these features. Of the descriptors used in the rule of 5 [39,40], the most of these compounds show few violations of the rule, and these are correlated with the results of gastrointestinal absorption.

The fact that most of the molecules present little affinity for P-gp is appropriate, since this favors drug absorption, and this protein has been associated with decreased bioavailability [41]. On the other hand, only some molecules seem to have the possibility of penetrating the blood–brain barrier, such as Urolithin A, Flavone, Resveratrol, and Chysin, which is interesting, as there is the possibility that they modulate the antinociceptive effects at the central level.

Additionally, in the future it would be interesting to perform the chemical characterization of the three varieties to elucidate which compounds are the differential effect, by their presence or by the proportion of its content, and to be able to associate them with the analgesia found. It is also advisable to isolate molecules present in the pomegranate that have not yet been evaluated in nociception in order to contribute to the elucidation of the mechanisms of action involved. To this end, the effect in the presence of antagonists should also be evaluated, assessing different routes of administration and the effect in different models and types of pain.

As demonstrated in this research, of the reported molecules present in *Punica granatum* L., the analysis shows that, in general, they could be considered safe. In addition, several studies have shown that pomegranate peel and juice do not have toxic effects even when administered in mega doses [42,43,44,45], so their consumption is considered safe, and these studies could be scaled up to a clinical level.

The results obtained set a precedent for the adequate choice of pomegranate varieties and the search for the best variety to counteract nociceptive behavior.

## 4. Materials and Methods

### 4.1. Extraction and Lyophilization Procedure

#### 4.1.1. Plants

Samples of three varieties of pomegranate (*Punica granatum* L.), whose common name in Mexico are bittersweet, green, and red, were obtained from the Valle del Mezquital region, in Tasquillo, Hidalgo, Mexico, in 2018. The fruits were collected at their commercial maturity stage by simple random probabilistic sampling, ensuring similar weight, color, and size. The taxonomic identification of the specimens was carried out in the Academic Area of Biology of the Autonomous University of the State of Hidalgo (voucher number: 733-9).

#### 4.1.2. Extraction

The peel (pericarp) and the juice of the three varieties of pomegranate were used. The peels were manually separated from the rest of the fruit, cut into squares of 0.25 to 0.5 cm, and dried in a darkroom at room temperature. Once dry, it was weighed, and a solid–liquid extraction was carried out by maceration with absolute ethanol (J.T. Baker, Ciudad de México, CDMX, Mexico), for 21 days (100 g of peel per 400 mL of solvent). Subsequently, rotoevaporation was carried out at reduced pressure (Büchi R200, Flawil, Switzerland) under the following conditions: bath temperature 30–35 °C; pressure 175 mbar; moderate rotation. The extraction percentages obtained were: PEx1 = 7.11, PEx2 = 10.48, and PEx3 = 12.13%. The final product was stored in a cool, dry environment until use.

#### 4.1.3. Lyophilization

The juice was extracted by manual pressing, then filtered from the seed and the carpel membrane that divides the arils. Subsequently, partial aqueous extraction was carried out by rotoevaporation at reduced pressure under the following conditions: bath temper-ature of 35–40 °C; pressure 70 mbar; fast rotation. A product reduced in water was obtained, which was taken to lyophilization (Labconco Freezone 4.5, Kansas City, MO, USA) at a temper-ature of −40 °C and 190 × 10^−3^ mBar. The result was weighed (The extraction percentages obtained were: Lj1 = 6.8; Lj2 = 6.98 and Lj3 = 7.1%). Lyophilized juices were hermetically stored in a cool and dry environment until use.

### 4.2. Reagents

Absolute ethanol (J.T. Baker, CDMX, Mexico) was used for maceration. For the pain test (2% formalin), 37% formaldehyde (J.T. Baker, CDMX, Mexico) in saline solution (PiSA, CDMX, Mexico) was used, and 1% Tween 80 (Sigma Aldrich, Burlington, MA, USA) was used as vehicle. Additionally, 0.5% Carboxymethylcellulose (Sigma Aldrich, Burlington, MA, USA) was used as the vehicle for ASA (Sigma Aldrich, Burlington, MA, USA).

The reference drug, extracts, and lyophilized juices were administered intragastrically on the same day they were prepared, and the administration volume for all animals was 4 mL/kg of weight.

### 4.3. Animals

International and national ethical aspects were addressed [46,47], Forty-eight male Wistar rats (*Rattus norvegicus*) weighing 180–200 g were used. Before the experiments, the animals were kept in an isolated room of the vivarium, at a temperature of 27 ± 2 °C and light/dark cycles of 12 h. The rats received sterile water and FormuLab Diet 5008 commercial food (LabDiet, St. Louis, MO, USA) ad libitum. Prior to the experiments, the rats underwent a fasting period of 8 h. The experiments were carried out in the research area of the vivarium of the Institute of Health Sciences of the Autonomous University of the State of Hidalgo. Maximum efforts were made to minimize the suffering of the animals and reduce their number to a minimum per group. At the end of the experiments, the animals were sacrificed in a CO_2_ euthanasia chamber.

### 4.4. Pain Test Procedure

The antinociceptive activity of the extracts and juices was evaluated using the formalin test. After fasting for 7 h, each rat was placed in a cylindrical-shaped acrylic cage for 1 h to adapt to the conditions. Mirrors with 45° angles were placed behind the cages to achieve peripheral visibility of the rats. In this test, the number of flinches of the right hind paw was measured for one hour (1 min every 5), after a subcutaneous injection (30 G gauge) of 50 µL of 2% formalin in the dorsal surface of the same paw, 30 min after the intragastric administration of the drug to be evaluated.

The study groups (*n* = 6) were: (1) Vehicle (Tween 80 at 1%), (2) ASA (316 mg/kg), groups 3 to 5, corresponded to: bittersweet pomegranate peel extract = PEx1, green pomegranate peel extract = PEx2, and red pomegranate peel extract = PEx3. Groups 6 to 8 corresponded to: bittersweet pomegranate lyophilized juice: Lj1; green pomegranate lyophilized juice: Lj2 and red pomegranate lyophilized juice: Lj3. All experimental groups received 316 mg/kg intragastric.

#### Measurement of Nociceptive Behavior in the Formalin Test

Paw flinching is one of the behaviors related to nociception in the formalin model and is characterized by spontaneous, rapid, brief flinching, or lifting of the leg. Therefore, each episode of shaking, vibrating, or raising the leg was counted as one flinch. The antinociceptive response was measured by evaluating the time course. Subsequently, the area under the curve was calculated by the trapezoid method [48] of the number of flinches in relation to time. From these results, the percentage of antinociception was calculated, using the formula presented below:% Antinociception = (AUC vehicle − AUC post-treatment/AUC vehicle) × 100(1)
where:

AUC Vehicle: Area under the curve of the vehicle group

AUC post-treatment: Area under the curve of the treated groups

Two types of pain are measured in this test: 0 to 10 min (phase 1) represents nociceptive pain, and 15 to 60 min (phase 2) corresponds to inflammatory pain.

### 4.5. In Silico Analysis

To obtain computational predictions, the simplified molecular input line entry system code (SMILES) of the main metabolites described for *Punica granatum* L. was used. The SMILES code was obtained from PubChem. The PASSonline platform was used to identify the biological effects and mechanisms associated with each molecule [49], and SwissADME was used for determinations of biopharmaceutical parameters [50]; pkCSM and admetSAR were used to identify possible risks linked to the compounds [51].

### 4.6. Statistical Analysis

The data obtained were expressed as the mean ± SE of each variable. The difference between means for each group was estimated by a one-way analysis of variance (ANOVA) with a Bonferroni posttest to establish the difference in means with a significance level of *p* < 0.05. Some data were treated by Student’s *t*-test (*p* < 0.05). Statistical analysis was performed with Graphpad Prism 6 for Windows software (GraphPad Software, San Diego, CA, USA) and Microsoft 365 Excel 2019 (Microsoft, Redmond, WA, USA).

## 5. Conclusions

The ethanolic extracts of peel and the lyophilized juices of three varieties of pomegranate from Mexico have an antinociceptive effect in a model of nociceptive and inflammatory pain with differences in efficacy between them, which may be a consequence of the different concentrations of their bioactive compounds present in each variety or the synergism of them. The lyophilized juice and ethanolic extract of green pomegranate peel has a greater antinociceptive effect than the other varieties in the formalin test. In recent years, the demand for functional food and beverages based on fruits rich in phytonutrients has increased to improve nutrition and health. In this sense, PEx and Lj can be considered as a natural alternative for the treatment of nociceptive and inflammatory pain.

## Figures and Tables

**Figure 1 plants-12-00131-f001:**
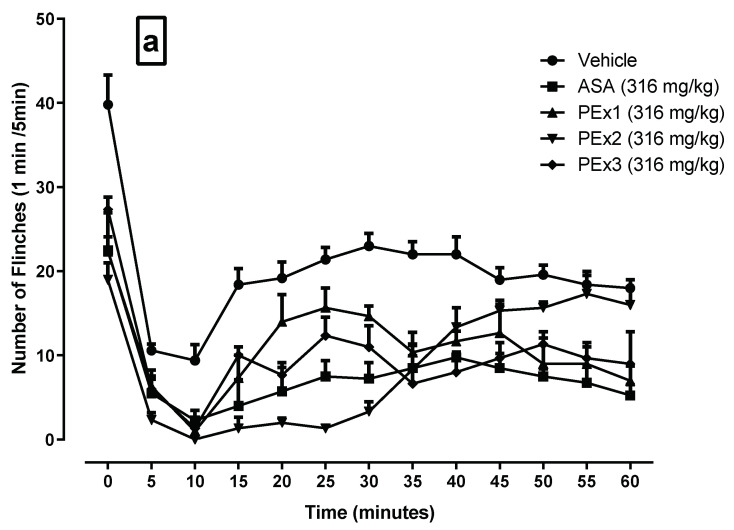
Time courses of the formalin test obtained by the intragastric administration of the extracts of peel and lyophilized juice of the three varieties of pomegranate. (**a**,**b**) shows the flinches by time intervals (1 min every 5 min, up to 60 min).

**Figure 2 plants-12-00131-f002:**
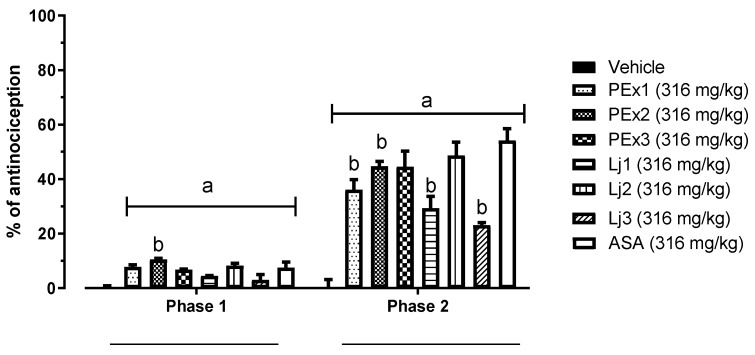
Percentage of antinociception of the different study groups by phases in the formalin test. a = Statistically significant difference versus Vehicle (*p* < 0.05), b = Statistically significant difference versus ASA (*p* < 0.05).

**Figure 3 plants-12-00131-f003:**
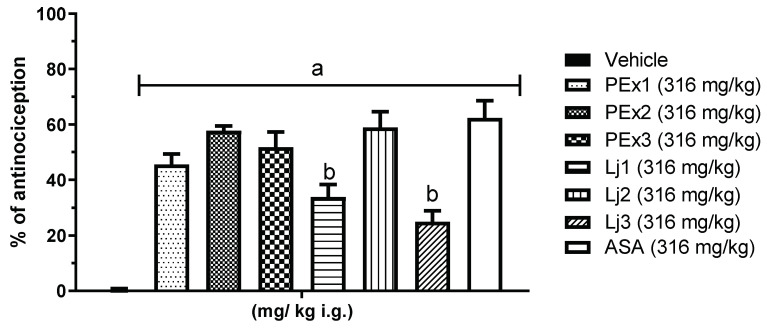
Percentage of global antinociception of the different treatment groups in the formalin test. a = Statistically significant difference versus Vehicle (*p* < 0.05), b = Statistically significant difference versus ASA (*p* < 0.05).

**Table 1 plants-12-00131-t001:** Pharmacological activities predicted for major compounds by PASSonline.

Compound	Antinociceptive		Anti-inflammatory		Antioxidant	
	Pa	Pi	Pa	Pi	Pa	Pi
*Punicalagin*	--	--	0.983	0.004	0.957	0.002
*Punicalin*	--	--	0.829	0.005	0.873	0.003
*Granatin B*	--	--	0.572	0.038	0.483	0.007
*Urolithin A*	0.326	0.163	0.572	0.038	0.888	0.003
*Corilagin*	--	--	0.700	0.016	0.671	0.004
*Flavone*	0.399	0.110	0.547	0.044	0.469	0.008
*Resveratrol*	0.503	0.034	0.554	0.042	0.546	0.005
*Quercetin*	0.362	0.137	0.689	0.017	0.872	0.003
*Apigenin*	0.348	0.147	0.644	0.008	0.732	0.004
*Chrysin*	0.341	0.152	0.637	0.025	0.708	0.004
*Galangin*	0.338	0.154	0.689	0.017	0.853	0.006
*Kaempferol*	0.345	0.149	0.676	0.008	0.856	0.002
*Ellagic acid*	0.474	0.053	0.759	0.010	0.699	0.004
*Gallic acid*	0.539	0.019	0.548	0.044	0.520	0.006
*Punicic acid*	0.540	0.018	0.675	0.019	0.341	0.018

Pa = probability to be active; Pi = probability to be inactive.

**Table 2 plants-12-00131-t002:** Molecular mechanism predicted for major compounds by PASSonline.

Compound	PKA Inhibitor		PKC Inhibitor		NOS2 Inhibitor		Histamine Release Inhibitor		TNF Expression Inhibitor	
	Pa	Pi	Pa	Pi	Pa	Pi	Pa	Pi	Pa	Pi
*Punicalagin*	0.934	0.001	0.558	0.003	0.289	0.005	--	--	--	--
*Punicalin*	0.801	0.002	0.248	0.003	0.253	0.075	--	--	--	--
*Granatin B*	0.995	0.000	0.336	0.003	0.240	0.085	--	--	--	--
*Urolithin A*	--	--	--	--	0.581	0.004	0.764	0.004	--	--
*Corilagin*	0.884	0.002	0.3310	0005	0.240	0.085	--	--	--	--
*Flavone*	--	--	--	--	0.565	0.005	0.732	0.004	0.461	0.008
*Resveratrol*	--	--	--	--	0.603	0.004	0.526	0.027	0.654	0.009
*Quercetin*	--	--	--	--	0.850	0.002	0.751	0.003	0.501	0.029
*Apigenin*	--	--	--	--	0.732	0.002	0.791	0.003	0.609	0.012
*Chrysin*	--	--	--	--	0.705	0.003	0.788	0.004	0.573	0.007
*Galangin*	--	--	--	--	0.784	0.002	0.690	0.005	0.449	0.042
*Kaempferol*	--	--	--	--	0.797	0.002	0.692	0.005	--	--
*Ellagic acid*	0.343	0.004	--	--	0.324	0.011	0.519	0.008	--	--
*Gallic acid*	0.324	0.004	--	--	0.382	0.022	0.654	0.006	0.560	0.018
*Punicic acid*	--	--	--	--	--	--	0.575	0.014	0.751	0.005

Pa = probability to be active; Pi = probability to be inactive.

**Table 3 plants-12-00131-t003:** Physicochemical properties of the main phytochemicals by SwissADME.

Compound/Property	LogP	TPSA	MW	HBD/HBA	RB	N Violations	PAINS
*Punicalagin*	0.07	518.76	1084.72	17/30	0	3	1
*Punicalin*	−0.83	385.24	782.53	13/22	0	3	1
*Granatin B*	−1.49	450.25	952.64	14/27	3	3	1
*Urolithin A*	2.06	70.67	228.20	2/4	0	0	0
*Corilagin*	−0.78	310.66	634.45	11/18	3	3	1
*Flavone*	3.18	30.21	222.24	0/2	1	0	0
*Resveratrol*	2.48	60.69	228.24	3/3	2	0	0
*Quercetin*	1.23	131.36	302.24	5/7	1	0	1
*Apigenin*	2.11	90.90	270.24	3/5	1	0	0
*Chrysin*	2.55	70.67	254.24	2/4	1	0	0
*Galangin*	1.99	90.90	270.24	3/5	1	0	0
*Kaempferol*	1.58	111.13	286.24	4/6	1	0	0
*Ellagic acid*	1.00	141.34	302.19	4/8	0	0	1
*Gallic acid*	0.21	97.99	170.12	4/5	1	0	0
*Punicic acid*	0.21	278.43	278.43	1/2	13	0	1

LogP = lipophilicity; TPSA = total polar surface area (Reference 100–150 Å); MW = molecular weight (reference ≤ 500 uma); HBD = hydrogen bond donors (reference ≤ 5); HBA = hydrogen bond acceptors (Reference ≤ 10); RB = number of rotatable bonds; n violations = n violations of Lipinski’s Rule of Five; PAINS = Pan assay interferences structures.

**Table 4 plants-12-00131-t004:** Absorption and distribution properties of the main phytochemicals by SwissADME.

Compound/Property	BBB Permeant	P-gp Substrate	GI Absorption
*Punicalagin*	No	Yes	Low
*Punicalin*	No	Yes	Low
*Granatin B*	No	Yes	Low
*Urolithin A*	Yes	No	High
*Corilagin*	No	Yes	Low
*Flavone*	Yes	No	High
*Resveratrol*	Yes	No	High
*Quercetin*	No	No	High
*Apigenin*	No	No	High
*Chrysin*	Yes	No	High
*Galangin*	No	No	High
*Kaempferol*	No	No	High
*Ellagic acid*	No	No	High
*Gallic acid*	No	No	High
*Punicic acid*	No	No	High

BBB permeant = blood–brain permeant; P-gp = P-glycoprotein; GI = gastrointestinal.

**Table 5 plants-12-00131-t005:** Toxicity parameters predicted for major compounds by admetSAR.

Compound/Property	AMES Toxicity	hERG	Carcinogenicity	Nephrotoxicity	Reproductive Toxicity	Oral Rat Acute Toxicity (LD_50_, mol/kg)	Acute Oral Toxicity (Category OCDE)
*Punicalagin*	**+** (0510)	**+** (0.763)	**−** (1.000)	**−** (0.844)	**+** (0.744)	2.74	III
*Punicalin*	**+** (0.540)	**+** (0.723)	**−** (0.985)	**−** (0.827)	**+** (0.744)	2.74	III
*Granatin B*	**+** (0.630)	**+** (0.690)	**−** (0.960)	**−** (0.860)	**+** (0.777)	3.01	III
*Urolithin A*	**−** (0.610)	**−** (0.900)	**−** (1.000)	**−** (0.697)	**+** (0.577)	2.48	III
*Corilagin*	**−** (0.500)	**+** (0.756)	**−** (0.942)	**−** (0.877)	**+** (0.766)	2.28	III
*Flavone*	**−** (0.670)	**−** (0.708)	**−** (0.888)	**−** (0.579)	**+** (0.566)	2.84	III
*Resveratrol*	**−** (0.820)	**−** (0.836)	**−** (0.530)	**+** (0.537)	**−** (0.555)	2.37	III
*Quercetin*	**+** (0900)	**−** (0.841)	**−** (1.000)	**−** (0.816)	**+** (0.766)	2.52	II
*Apigenin*	**−** (0.830)	**−** (0.897)	**−** (1.000)	**−** (0.663)	**+** (0.655)	2.69	III
*Chrysin*	**−** (0.985)	**−** (0.800)	**−** (1.000)	**−** (0.628)	**+** (0.655)	1.92	III
*Galangin*	**−** (0.790)	**−** (0.871)	**−** (1.000)	**−** (0.714)	**+** (0.766)	2.24	II
*Kaempferol*	**+** (0.730)	**−** (0.914)	**−** (1.000)	**−** (0.747)	**+** (0766)	1.83	II
*Ellagic acid*	**−** (0.820)	**−** (0.804)	**−** (1.000)	**−** (0.844)	**+** (0.633)	1.71	II
*Gallic acid*	**−** (0.950)	**−** (0.842)	**−** (0.662)	**−** (0.697)	**+** (0.677)	2.31	III
*Punicic acid*	**−** (0.879)	**−** (0.421)	**−** (0.671)	**−** (0.589)	**−** (0.791)	1.53	IV

(**+**) Presence of toxicological effect; (**−**) no toxicological effect followed by probability; green, satisfactory; yellow, intermediate satisfaction; Red, unsatisfactory. Acute Oral Toxicity (Category I contains compounds with LD_50_ values less than or equal to 50 mg/kg. Category II contains compounds with LD_50_ values greater than 50 mg/kg but less than 500 mg/kg. Category III includes compounds with LD_50_ values greater than 500 mg/kg but less than 5000 mg/kg. Category IV consisted of compounds with LD_50_ values greater than 5000 mg/kg).

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
