# Peer review of "Differential Antinociceptive Efficacy of Peel Extracts and Lyophilized Juices of Three Varieties of Mexican Pomegranate (Punica granatum L.) in the Formalin Test"

_plants, 2022, doi:10.3390/plants12010131_

Round 1

Reviewer 1 Report

Thank you for sending your manuscripts to Plant 

the manuscript is well written, the methods targeted the research design 

however my only comment is that the biological part of the paper is too small (and based only on the formalin test) 

The authors assumed the postulated mechanism of actions for the phytoactive constituents based on literature without performing any experiments to measure the levels of PKA or PKC, iNOS, histamine or tumor necrosis factor or at least some of them to confirm his hypothesis 

I strongly recommend measuring these biological markers to improve the quality of your manuscript

Author Response

14th December 2022

Editor-in-Chief

Plants

Thank you for your decision and constructive comments on our manuscript “Differential Antinociceptive Efficacy of Peel Extracts and Lyophilized Juices of Three Varieties of Mexican Pomegranate (Punica granatum L.) in the formalin test” by José Antonio Guerrero-Solano, Mirandeli Bautista, Josué Vidal Espinosa-Juárez, Luis Alfonso Moreno-Rocha, Gabriel Betanzos-Cabrera, Liana Claudia Salanță , Minarda De la O Arciniega, Elena G. Olvera-Hernández and Osmar Antonio Jaramillo-Morales for possible publication in plants. We appreciate the time and effort that you and the reviewers have dedicated to providing the manuscript. We have been able to incorporate changes to reflect all the suggestions provided by the reviewers. Please find below the response. It is pertinent to mention that the changes have been highlighted for track changes in the revised manuscript.

We would like to thank you and the reviewers for your very helpful comments.

Sincerely yours,

Osmar Antonio Jaramillo-Morales, PhD.

Reviewer 1:

The manuscript is well written, the methods targeted the research design 

however my only comment is that the biological part of the paper is too small (and based only on the formalin test) 

The authors assumed the postulated mechanism of actions for the phytoactive constituents based on literature without performing any experiments to measure the levels of PKA or PKC, iNOS, histamine or tumor necrosis factor or at least some of them to confirm his hypothesis 

I strongly recommend measuring these biological markers to improve the quality of your manuscript

Response: We appreciate your comment regarding our work, of course, derived from these results, a large number of questions arise that will need to be answered, but since the main idea of the work was to demonstrate the antinociceptive effects, the determination of biological markers were not considered, however, his comment is very valuable to address it in the future, this observation was taken up again in the discussion.

Reviewer 2 Report

Authors studied antinociceptive effect of mexican pomegranate peel extract and lyophilized juice of bittersweet, green and red pomegranate using formalin test as a model of acute pain and toxicity of pomegranate compounds. They found that the lyophilized juice and ethanolic extract of green pomegranate peel have antinociceptive effect in the formalin test and they may be used as a natural alternative for the treatment of nociceptive and inflammatory pain. 

The work is interesting, methods and results are clear, well discussed. Looking for active compounds in plants to treat various health conditions is a good direction in research. 

Author Response

14th December 2022

Editor-in-Chief

Plants

Thank you for your decision and constructive comments on our manuscript “Differential Antinociceptive Efficacy of Peel Extracts and Lyophilized Juices of Three Varieties of Mexican Pomegranate (Punica granatum L.) in the formalin test” by José Antonio Guerrero-Solano, Mirandeli Bautista, Josué Vidal Espinosa-Juárez, Luis Alfonso Moreno-Rocha, Gabriel Betanzos-Cabrera, Liana Claudia Salanță , Minarda De la O Arciniega, Elena G. Olvera-Hernández and Osmar Antonio Jaramillo-Morales for possible publication in plants. We appreciate the time and effort that you and the reviewers have dedicated to providing the manuscript. We have been able to incorporate changes to reflect all the suggestions provided by the reviewers. Please find below the response. It is pertinent to mention that the changes have been highlighted for track changes in the revised manuscript.

We would like to thank you and the reviewers for your very helpful comments.

Sincerely yours,

Osmar Antonio Jaramillo-Morales, PhD.

Reviewer 2:

Authors studied antinociceptive effect of mexican pomegranate peel extract and lyophilized juice of bittersweet, green and red pomegranate using formalin test as a model of acute pain and toxicity of pomegranate compounds. They found that the lyophilized juice and ethanolic extract of green pomegranate peel have antinociceptive effect in the formalin test and they may be used as a natural alternative for the treatment of nociceptive and inflammatory pain. 

The work is interesting, methods and results are clear, well discussed. Looking for active compounds in plants to treat various health conditions is a good direction in research. 

Response: We very much appreciate your comment regarding our work.

Reviewer 3 Report

The study reported an antinociceptive effect from the ethanolic extracts of peel and the lyophilized juices. The preliminary data is inspiring, however, more detailed study is necessary to clarify which compound(s) is responsible for this effect. The questions below may also need further consideration.

·         Why not combine 1a and 1b, since they are expressing the same measurement

·         Why used lyophilized juice instead of concentrated?

·         Fig 3, Why there are different effects on antinociception from different extracts and juices? Why Lj2 is much better than Lj1 and 3?

·         What is the Peel extract and juice Composition

Author Response

14th December 2022

Editor-in-Chief

Plants

Thank you for your decision and constructive comments on our manuscript “Differential Antinociceptive Efficacy of Peel Extracts and Lyophilized Juices of Three Varieties of Mexican Pomegranate (Punica granatum L.) in the formalin test” by José Antonio Guerrero-Solano, Mirandeli Bautista, Josué Vidal Espinosa-Juárez, Luis Alfonso Moreno-Rocha, Gabriel Betanzos-Cabrera, Liana Claudia Salanță , Minarda De la O Arciniega, Elena G. Olvera-Hernández and Osmar Antonio Jaramillo-Morales for possible publication in plants. We appreciate the time and effort that you and the reviewers have dedicated to providing the manuscript. We have been able to incorporate changes to reflect all the suggestions provided by the reviewers. Please find below the response. It is pertinent to mention that the changes have been highlighted for track changes in the revised manuscript.

We would like to thank you and the reviewers for your very helpful comments.

Sincerely yours,

Osmar Antonio Jaramillo-Morales, PhD.

Reviewer 3:

The study reported an antinociceptive effect from the ethanolic extracts of peel and the lyophilized juices. The preliminary data is inspiring; however, more detailed study is necessary to clarify which compound(s) is responsible for this effect. The questions below may also need further consideration.

  • Why not combine 1a and 1b, since they are expressing the same measurement:

Response: We appreciate the observation and if we consider combining both figures, however we decided not to do so in order not to saturate the image, since each figure shows a total of three doses

  • Why used lyophilized juice instead of concentrated?

Response: Lyophilization process maximizes product stability, maintains biological function, and enables easier storage. Also reduce the risk of sample contamination. For the experiments, lyophilized juice can be weighed more accurately for administration.

  • Fig 3, Why there are different effects on antinociception from different extracts and juices? Why Lj2 is much better than Lj1 and 3?

Response: The variation in the effects of the extract and the juice may be due to the variation in the type and quantity of components present in each of them, this is explained in the discussion. Regarding the effect obtained with Lj, it is interesting and could be explained by the amount of metabolites present in each dose used, it is important to note that since there is a wide variety of components, positive interactions can occur in some, generating potentiation, however negative interactions can also occur, such as antagonism, for this reason, one possibility is that in Lj2 metabolites are present in the correct quantity that generate summation effects with respect to Lj1, however, by continuing to increase the doses as in Lj3, it is possible that the necessary amount of a metabolite that blocks or counteracts the antinociceptive effect is already reached, this is detailed in the discussion.

  • What is the Peel extract and juice Composition

Response: This observation is interesting; however, the main objective of this research was to show whether the juice and the extract had antinociceptive effects, however we believe that in the future it would be relevant to show the type of components present and which would be directly responsible for the observed effects, or if it is the combination of them. This is described in the discussion.

Reviewer 4 Report

This study aimed to evaluate the antinociceptive effect of three Mexican varieties of pomegranate, using the formalin assay as an in vivo model. The results showed that of the three ethanolic extracts of the skin and of the lyophilized ones of the juice, the most active with respect to the antinociceptive effect corresponded to species 2 (green pomegranate). In addition, the possible mechanisms of action that the extracts and freeze-dried products could have to reduce the inflammatory response that occurs in phase two of the formalin model were demonstrated through in silico studies.

Despite the fact that the work carried out is interesting, it is suggested to enrich the study with more in vivo and/or in vitro experiments through which it is possible to elucidate the mechanism(s) of action of the bioactive compounds both in the ethanolic extract of the skin, as of the freeze-dried juice, on the antinociceptive effect found. In addition, it is suggested to carry out more in silico studies to demonstrate the interaction of the bioactive compounds mainly responsible for the antinociceptive and anti-inflammatory activity of the green pomegranate variety with proteins associated with these processes (for example, COX, NOS, etc.) .

For example, if it has already been shown that this effect is mainly associated with the decrease in the inflammatory process produced in phase two of the formalin model, then:

1. Why not experimentally evaluate the expression or activity of proteins involved in this process, previously identifying the major bioactive compounds? This would significantly increase the quality of the research conducted.

2. Is there a history of the interactions that occur between some of the bioactive compounds and proteins mentioned through in silico studies? Otherwise, it would be recommendable to carry out the corresponding analysis using the molecular coupling method (docking). Furthermore, it would be important to determine the physicochemical and pharmacological properties using different online servers.

Author Response

14th December 2022

Editor-in-Chief

Plants

Thank you for your decision and constructive comments on our manuscript “Differential Antinociceptive Efficacy of Peel Extracts and Lyophilized Juices of Three Varieties of Mexican Pomegranate (Punica granatum L.) in the formalin test” by José Antonio Guerrero-Solano, Mirandeli Bautista, Josué Vidal Espinosa-Juárez, Luis Alfonso Moreno-Rocha, Gabriel Betanzos-Cabrera, Liana Claudia Salanță , Minarda De la O Arciniega, Elena G. Olvera-Hernández and Osmar Antonio Jaramillo-Morales for possible publication in plants. We appreciate the time and effort that you and the reviewers have dedicated to providing the manuscript. We have been able to incorporate changes to reflect all the suggestions provided by the reviewers. Please find below the response. It is pertinent to mention that the changes have been highlighted for track changes in the revised manuscript.

We would like to thank you and the reviewers for your very helpful comments.

Sincerely yours,

Osmar Antonio Jaramillo-Morales, PhD.

Reviewer 4:

This study aimed to evaluate the antinociceptive effect of three Mexican varieties of pomegranate, using the formalin assay as an in vivo model. The results showed that of the three ethanolic extracts of the skin and of the lyophilized ones of the juice, the most active with respect to the antinociceptive effect corresponded to species 2 (green pomegranate). In addition, the possible mechanisms of action that the extracts and freeze-dried products could have to reduce the inflammatory response that occurs in phase two of the formalin model were demonstrated through in silico studies.

Despite the fact that the work carried out is interesting, it is suggested to enrich the study with more in vivo and/or in vitro experiments through which it is possible to elucidate the mechanism(s) of action of the bioactive compounds both in the ethanolic extract of the skin, as of the freeze-dried juice, on the antinociceptive effect found. In addition, it is suggested to carry out more in silico studies to demonstrate the interaction of the bioactive compounds mainly responsible for the antinociceptive and anti-inflammatory activity of the green pomegranate variety with proteins associated with these processes (for example, COX, NOS, etc.).

For example, if it has already been shown that this effect is mainly associated with the decrease in the inflammatory process produced in phase two of the formalin model, then:

  1. Why not experimentally evaluate the expression or activity of proteins involved in this process, previously identifying the major bioactive compounds? This would significantly increase the quality of the research conducted.

Response: We value your comment, of course, derived from these, it is necessary in the future to deepen the associated mechanisms of action and identify the metabolites responsible for the effects. We express these perspectives in the discussion, due to the time they give us in the journal to generate the answer is not possible to carry out such experiments.

  1. Is there a history of the interactions that occur between some of the bioactive compounds and proteins mentioned through in silico studies? Otherwise, it would be recommendable to carry out the corresponding analysis using the molecular coupling method (docking). Furthermore, it would be important to determine the physicochemical and pharmacological properties using different online servers.

Response: We appreciate your observation, in this sense, there is evidence of the in vivo interaction of the molecules in Table 2 with the protein systems, and the references of these studies are in the paper.

Regarding the physicochemical and pharmacological properties, these were attached and are shown in Tables 3 and 4.

Round 2

Reviewer 1 Report

Thank you for your clarification

Please consider performing further experiments to elucidate the mechnism of the effective compounds in the future 

Author Response

20th December 2022

Editor-in-Chief

Plants

Thank you for your decision and constructive comments on our manuscript “Differential Antinociceptive Efficacy of Peel Extracts and Lyophilized Juices of Three Varieties of Mexican Pomegranate (Punica granatum L.) in the formalin test” by José Antonio Guerrero-Solano, Mirandeli Bautista, Josué Vidal Espinosa-Juárez, Luis Alfonso Moreno-Rocha, Gabriel Betanzos-Cabrera, Liana Claudia Salanță , Minarda De la O Arciniega, Elena G. Olvera-Hernández and Osmar Antonio Jaramillo-Morales for possible publication in plants. We appreciate the time and effort that you and the reviewers have dedicated to providing the manuscript. We have been able to incorporate changes to reflect all the suggestions provided by the reviewers. Please find below the response. It is pertinent to mention that the changes have been highlighted for track changes in the revised manuscript. We would like to thank you and the reviewers for your very helpful comments.

Sincerely yours,

Osmar Antonio Jaramillo-Morales, PhD.

Review 1

Thank you for your clarification

Please consider performing further experiments to elucidate the mechanism of the effective compounds in the future

Response: We appreciate your comment regarding our work, of course, derived from these results, a large number of questions arise that must be answered in the future.

Reviewer 3 Report

Further experiments are needed to confirm the effects, elucidate the mechnism and clarify the effective compounds

Author Response

20th December 2022

Editor-in-Chief

Plants

Thank you for your decision and constructive comments on our manuscript “Differential Antinociceptive Efficacy of Peel Extracts and Lyophilized Juices of Three Varieties of Mexican Pomegranate (Punica granatum L.) in the formalin test” by José Antonio Guerrero-Solano, Mirandeli Bautista, Josué Vidal Espinosa-Juárez, Luis Alfonso Moreno-Rocha, Gabriel Betanzos-Cabrera, Liana Claudia Salanță , Minarda De la O Arciniega, Elena G. Olvera-Hernández and Osmar Antonio Jaramillo-Morales for possible publication in plants. We appreciate the time and effort that you and the reviewers have dedicated to providing the manuscript. We have been able to incorporate changes to reflect all the suggestions provided by the reviewers. Please find below the response. It is pertinent to mention that the changes have been highlighted for track changes in the revised manuscript. We would like to thank you and the reviewers for your very helpful comments.

Sincerely yours,

Osmar Antonio Jaramillo-Morales, PhD.

Review 3

Further experiments are needed to confirm the effects, elucidate the mechnism and clarify the effective compounds.

Response: We value your comment, of course, derived from these, it is necessary in the future to deepen the associated mechanisms of action and identify the metabolites responsible for the effects. We express these perspectives in the discussion, due to the time they give us in the journal to generate the answer is not possible to carry out such experiments.

Reviewer 4 Report

Based on the observations made previously, the authors have answered these. However, it is suggested:

Answer to observation 1:

A reasonable answer was given from the point of view that the experimental evaluation of the possible targets involved in the antinociceptive effect of the compounds evaluated will be carried out in future works, considering that for now there is not enough time to carry out such experiments. .

Answer to observation 2:

Despite the response that Table 2 shows the possible molecular mechanisms involved, the meaning of the numerical values shown in said table (Pa and Pi) is not clearly explained. In addition, the response mentions that references are included, but does not specifically indicate what they are, nor does it adequately justify why the authors do not consider it necessary to carry out a docking study with the possible targets involved.

In addition, the authors have included Tables 3 and 4, but they do not describe the meaning of each of the parameters predicted by the online servers, nor do they correlate it with its importance at a biological level, since they only do so with some of them. They also mention that the compounds comply with Lipinski's rule of 5, but do not delve into what the latter consists of.

Author Response

20th December 2022

Editor-in-Chief

Plants

Thank you for your decision and constructive comments on our manuscript “Differential Antinociceptive Efficacy of Peel Extracts and Lyophilized Juices of Three Varieties of Mexican Pomegranate (Punica granatum L.) in the formalin test” by José Antonio Guerrero-Solano, Mirandeli Bautista, Josué Vidal Espinosa-Juárez, Luis Alfonso Moreno-Rocha, Gabriel Betanzos-Cabrera, Liana Claudia Salanță , Minarda De la O Arciniega, Elena G. Olvera-Hernández and Osmar Antonio Jaramillo-Morales for possible publication in plants. We appreciate the time and effort that you and the reviewers have dedicated to providing the manuscript. We have been able to incorporate changes to reflect all the suggestions provided by the reviewers. Please find below the response. It is pertinent to mention that the changes have been highlighted for track changes in the revised manuscript. We would like to thank you and the reviewers for your very helpful comments.

Sincerely yours,

Osmar Antonio Jaramillo-Morales, PhD.

Review 4

Answer to observation 2:

Despite the response that Table 2 shows the possible molecular mechanisms involved, the meaning of the numerical values shown in said table (Pa and Pi) is not clearly explained. In addition, the response mentions that references are included, but does not specifically indicate what they are, nor does it adequately justify why the authors do not consider it necessary to carry out a docking study with the possible targets involved.

Response: We appreciate your valuable comment that undoubtedly raises the quality of our work, in this sense we have added an explanation of the meaning of Pa and Pi on page 5, lines 147-149

Regarding the references that mention the evidence of the interaction of the bioactive components and the target proteins, they correspond to 33-38 of the list.

Regarding Docking, we do not carry them out because despite the fact that the results shown in Table 2 are based on the structure-activity relationship, and there are some with great potential to be active, there is a possibility that the effects are not generated directly on the target but rather on the signaling pathway, so we believe that we would have to carry out in vivo validation in the next phase and then implement the molecular docking assays in a more directed way on the targets involved.

In addition, the authors have included Tables 3 and 4, but they do not describe the meaning of each of the parameters predicted by the online servers, nor do they correlate it with its importance at a biological level, since they only do so with some of them. They also mention that the compounds comply with Lipinski's rule of 5, but do not delve into what the latter consists of

Response: We appreciate your comments, in this regard, we have described the meaning of each parameter in tables 3 and 4 in the results section (pages 6 and 7), likewise we have made the correlation of the biological importance in the discussion (pages 9-10, lines 300-311).